# Ribosome Biogenesis and Function in Cancer: From Mechanisms to Therapy

**DOI:** 10.3390/cancers17152534

**Published:** 2025-07-31

**Authors:** Kezia Gitareja, Shalini S. Chelliah, Elaine Sanij, Shahneen Sandhu, Jian Kang, Amit Khot

**Affiliations:** 1St. Vincent’s Institute of Medical Research, Melbourne, VIC 3065, Australia; kgitareja@svi.edu.au (K.G.); sschelliah@svi.edu.au (S.S.C.); esanij@svi.edu.au (E.S.); 2Department of Medicine—St. Vincent’s Hospital, University of Melbourne, Melbourne, VIC 3065, Australia; 3Department of Biochemistry and Molecular Biology, Monash University, Melbourne, VIC 3800, Australia; 4Cancer Research Division, Peter MacCallum Cancer Centre, Melbourne, VIC 3000, Australia; shahneen.sandhu@petermac.org; 5Sir Peter MacCallum Department of Oncology, University of Melbourne, Melbourne, VIC 3000, Australia; 6Clinical Haematology, Peter MacCallum Cancer Centre, Melbourne, VIC 3000, Australia

**Keywords:** ribosome biogenesis, mRNA translation, nucleolus, pol I transcription, cancer therapy

## Abstract

The ribosome is a cellular structure that supports protein production through the process of mRNA translation. Ribosome biogenesis is an intricate process responsible for the synthesis of these machineries. Cancer cells exploit altered ribosome biogenesis and mRNA translation to sustain cell growth and proliferation. This presents a key vulnerability that can be targeted in cancer therapy. In this review, we discuss dysregulation of ribosome biogenesis and mRNA translation in cancer and assess current strategies investigating the inhibition of these processes.

## 1. Introduction

The significance of ribosome biogenesis and mRNA translation in promoting tumorigenesis has been increasingly recognised in the last few decades. Nucleolar morphology is a well-established hallmark of cancer and has been recognised since the early 20th century as a diagnostic indicator of malignancy [1,2,3,4]. As oncogenic signalling pathways like MAPK/ERK, PI3K/Akt/mTOR, and others were uncovered, the link between faulty ribosome function and cancer became more apparent [5,6]. These oncogenic pathways cooperate to enhance ribosome and protein synthesis, promoting “translation addiction” in cancer cells. In fact, mutations in tumour suppressor genes and oncogenes, which are present across all cancer types, affect ribosome biogenesis and mRNA translation [7,8]. Individuals with ribosomopathies who have an increased risk of developing cancers also provide direct evidence that defects in the translation machinery contribute to oncogenesis [9,10].

In this review, we outline the process of ribosome biogenesis and mRNA translation and how their dysregulation contributes to cancer, highlight the importance of the nucleolus and the nucleolar stress response in guarding against cancer progression, and discuss promising ribosome-targeting strategies for cancer therapy.

## 2. Ribosome Synthesis and Function

The production of eukaryotic ribosomes is one of the most intricate biological processes. It requires the precise coordination of all three RNA polymerases (Pol I, II, and III) with numerous auxiliary factors to ensure accurate transcription and synthesis of ribosomal components and the assembly of precursor ribosomal particles [11,12,13]. Ribosome biogenesis occurs in the nucleolus, a specialised nuclear compartment that forms around nucleolar organiser regions of the chromatin containing repeated sequences of 47S ribosomal RNA (rRNA) genes [14,15]. These rRNA genes are clustered on the short arms of acrocentric chromosomes 13, 14, 15, 21, and 22, typically arranged in a head-to-tail, tandem configuration [16,17]. A canonical rRNA gene unit is approximately 43 kilobases, consisting of a 47S precursor rRNA (pre-rRNA) coding region, flanked upstream by a regulatory promoter and repetitive enhancer elements, and downstream by a transcription terminator sequence, embedded within an intergenic spacer (IGS) [18]. The 47S pre-rRNA harbours sequences encoding 18S, 5.8S, and 28S rRNAs, separated by 5′ and 3′ external transcribed spacers (ETS) and two internal transcribed spacers (ITS1, ITS2) (Figure 1). Whilst rRNA genes are described as tandem repeats, studies have revealed the presence of several palindromic sequences and multiple rRNA gene variants that are differentially expressed [19]. Notably, in the first complete human genome assembly, it was revealed that the rRNA gene units, which were originally assumed highly homogenous, contain sequence variants [20]. The rRNA gene repeats exhibit chromosome-specific patterns, differing systematically across chromosomes, and to a lesser extent, they include variability between repeats within each array [20,21]. These variants contain single nucleotide changes, insertions and deletions, and large structural differences in the IGS and coding regions, all of which may contribute to ribosome heterogeneity [20,21].

### 2.1. Transcription of rRNA Genes and Synthesis of RPs and RBFs

Each human cell contains up to 400 copies of 47S rRNA genes, though only a subset is actively transcribed at any given time [18,22,23]. Transcriptionally active rRNA genes exist in a euchromatin state, characterised by DNA hypomethylation and acetylated histones [24]. This state is maintained by the upstream binding factor (UBF), a key regulator of rRNA gene transcription [22,23,25]. UBF binds as a dimer via its high mobility group domains to the core promoter and upstream control element (UCE) of rRNA gene promoters, inducing substantial DNA topological changes that promote chromatin accessibility and the recruitment of Pol I (Figure 2). Another crucial component for the initiation of Pol I transcription is selectivity factor 1 (SL1), which specifically binds to the core promoter [26,27,28,29]. SL1 is composed of TATA-box-binding protein (TBP) and TBP-associated factors (TAFs), including TAF1A, TAF1B, TAF1C, TAF1D, and TAF12 [27,30]. SL1 cooperates with UBF at the rRNA gene promoter to form a pre-initiation complex (PIC) that ultimately facilitates Pol I recruitment [31,32]. Pol I is incorporated into the PIC through its interaction with Pol I-specific transcription factor RRN3, which acts as a bridging factor between SL1 and Pol I, allowing the formation of a transcriptionally competent Pol I machinery [31,32]. Topoisomerase IIα, which relieves supercoiling tension generated during Pol I transcription, is also a critical element of transcription initiation [33,34]. Together, the coordinated actions of transcription factors, enzymes, and other regulatory proteins within the nucleolus ensure efficient Pol I transcription to produce the 47S pre-rRNA (Figure 2).

At the same time, 5S rRNA is transcribed by Pol III in the nucleoplasm, where its 3′ end is subsequently trimmed by exonucleases [35,36]. The processed 5S rRNA assembles with two ribosomal proteins (RPs), RPL5 and RPL11, into a 5S ribonucleoprotein (RNP) complex, and is then transported into the nucleolus, where it is incorporated into ribosomal particles [37,38].

Other components involved in ribosome biogenesis include ribosome biogenesis factors (RBFs) and non-coding small nucleolar RNAs (snoRNAs). Over 200 snoRNAs have been identified in humans; they are synthesised in the nucleoplasm before being assembled into small nucleolar ribonucleoprotein (snoRNP) complexes [39,40,41]. These snoRNP complexes have a primary function in guiding site-specific rRNA cleavage and modifications [40,42,43]. Additionally, RBFs and RPs assist in rRNA processing, whilst playing key roles in ribosome assembly and remodelling [13]. There are several hundred RBFs functioning as chaperones, transiently associating with precursor ribosomal particles during ribosome biogenesis [44,45]. In contrast, RPs are integrated into the ribosome.

### 2.2. Pre-40S and -60S Ribosomal Subunits Processing and Modification

Nascent 47S pre-rRNA strands embedded with sequences of 18S, 5.8S and 28S rRNA are quickly bound by a subset of snoRNPs, RPs, and RBFs, forming a large 90S processome in the nucleolus [46,47,48]. These complexes undergo a series of orchestrated cleavages by endonucleases and exonucleases, sequentially removing the non-coding ETS and ITS regions from pre-rRNA molecules (Figure 2) [49,50].

Throughout the process of ribosome biogenesis, pre-rRNAs undergo extensive co-transcriptional modifications to assist with rRNA folding. Most of these modifications are catalysed by box C/D and H/ACA snoRNPs complexes [42,43,51]. Each snoRNP complex contain a 60–170 nucleotide snoRNA, that guides the complex to a specific pre-rRNA site, four structural proteins, and a dedicated modification enzyme [39,41]. Box C/D snoRNPs contain fibrillarin (FBL), a methyltransferase that catalyses 2′-O-methylation of the rRNA backbone, whilst box H/ACA snoRNPs consist of the pseudouridine synthase dyskerin (DKC1) which causes rRNA pseudouridylation [42,43]. Other types of modifications like base methylation and acetylation are introduced by independent enzymes [52,53,54]. These modifications are integral to ribosome subunit assembly, establishing a defined rRNA structure to prevent improper RNA annealing events and occurrence of premature formation of secondary structures. Notably, the cumulative loss of rRNA modifications disrupts ribosomal assembly and subsequent mRNA translation; this deregulation is widely implicated in cancer [55,56].

### 2.3. Nuclear Export of Pre-Ribosomal Particles and Assembly into a Mature 80S Ribosome

After the cleavage event in ITS1, the maturation of pre-40S and -60S ribosomal particles occurs separately, each involving a series of RBF-mediated alterations and modifications of rRNAs in the nucleus [49,50]. The 18S rRNA assembles into the pre-40S subunit along with RPs designated as RPS, whilst the pre-60S subunit contains 5S, 5.8S and 28S rRNAs with RPs designated as RPL [13]. Both the pre-40S and -60S ribosomal subunits, and their associated RBFs are then exported into the cytoplasm through the nuclear pore complex [57,58,59]. This nuclear-cytoplasmic export of pre-ribosomal subunits primarily depends on the nuclear export receptor exportin 1 (XPO1), also known as chromosomal maintenance 1 (CRM1) (Figure 2) [58]. In the final stages of ribosome biogenesis, RBFs which facilitate nuclear export and restrict early ribosomal subunit coupling and translation initiation are removed [49]. This results in formation of a 40S subunit, containing 33 RPs, and a 60S subunit, containing 47 RPs, in the cytoplasm. These ribosome subunits then assemble with translation factors into a functional 80S ribosome, competent for mRNA translation.

### 2.4. mRNA Translation

The process of mRNA translation can be divided into four main steps: initiation, elongation, termination, and ribosome recycling [60,61]. It relies on the interplay of ribosome subunits, translation factors termed as eukaryotic initiation factors (eIFs) and elongation factors (eEFs), amino acids, and transfer RNAs (tRNAs). In a rate-limiting initiation step, 40S ribosome subunit interacts with a complex, comprising of GTP-bound eIF2, initiator methionyl tRNAi (tRNAi^Met^), and translation factors eIF3, eIF1, eIF5, and eIF1A, to form a 43S PIC [60]. The 43S PIC associates with mRNA-bound eIF4F complex, composed of cap-binding protein eIF4E, RNA helicase eIF4A, and scaffolding protein eIF4G, creating a 48S translation initiation complex. Processed mRNA molecules associate with eIF4F by binding eIF4E at the 5′ 7-methylguanosine (m^7^G) cap and eIF4G at the 3′ polyadenylated tail [62,63]. The 48S complex scans the 5′ untranslated region (UTR) of mRNAs until an AUG start codon is recognised, prompting the recruitment of the 60S ribosome subunit. This results in multiple GTP hydrolysis reactions and the displacement of translation initiation factors from the complex, allowing the formation of the mature 80S ribosome and signalling for translation elongation.

During the elongation phase, amino acids are repeatedly attached to a growing polypeptide chain [62,64]. This process is directed by base-pairing mRNA codons with corresponding anticodons of tRNAs. eEF1A decodes the codons and delivers aminoacyl-tRNA to the ribosome, allowing formation of the peptide bond between amino acids [64]. Following formation of peptide bonds, eEF2 mediates ribosome translocation, triggering the recruitment of subsequent aminoacyl-tRNAs. Translation terminates when an in-frame stop codon (UAA, UAG or UGA) is detected, resulting in the release of tRNA and the dissociation of 40S and 60S ribosomal subunits from nascent proteins [65,66]. These ribosomal subunits are then recycled for translation of other mRNAs [66].

The process of ribosome biogenesis is highly coordinated and energy-consuming. It is estimated that there are 10 million ribosomes in a human cell, constituting up to 6% of total protein mass, to meet the demands of protein synthesis [67]. In the case of impaired ribosome biogenesis, cells must immediately halt cell cycle progression to prevent incomplete growth and division.

## 3. The Nucleolus and Nucleolar Stress Response

The nucleolus is a highly organised, membrane-less nuclear compartment that is essential for precise regulation of ribosome biogenesis [68,69,70,71,72]. Pol I transcription, rRNA processing, and ribosomal subunit assembly underpin nucleolar structure, which is maintained through protein–protein interactions within distinct compartments [9,73]. The nucleolus exhibits liquid-like properties, forming a distinct phase-separated domain [14,70,74]. Formation of the nucleolus is driven by Pol I transcription, during which rRNA and RNPs coalesce into phase-separated droplets without membranes [14,75,76,77]. Differential miscibility of individual RNPs gives rise to a dynamic ‘tripartite architecture’, which consists of three distinct immiscible layers forming a droplet-within-droplet architecture. This unique organisation, observable by electron or light microscopy, enables compartmentalisation of successive steps in ribosome biogenesis [14,70,78,79,80,81].

At the core of the nucleolus lie fibrillar centres (FCs), which contain the transcription machinery for Pol I and its components, such as UBF and RRN3 (Figure 3) [75,82]. FC also acts as a storage repository site for inactive rRNA gene repeats and unengaged Pol I transcription factors [75,82]. Surrounding the FCs, the dense fibrillar components (DFCs) host early pre-rRNA processing factors, such as FBL and nucleolin (NCL). Transcription of rRNA genes by Pol I takes place at the boundary of the FC and DFC [75,82]. The outermost layer is the granular component (GC), where late steps of ribosome biogenesis occur, such as the assembly of rRNA with RPs to form pre-40S and -60S ribosomal subunits [75,76,83,84].

Disruption of ribosome biogenesis through inhibition of rRNA gene transcription or defects in rRNA processing and ribosome formation can disrupt the nucleolar architecture, leading to disintegration of the nucleolus, release of sequestered proteins into the nucleoplasm or cytoplasm, and activation of stress signalling pathways, known as the nucleolar stress response [70,85,86,87,88]. Accordingly, the nucleolus acts as a control centre that integrates both intrinsic and extrinsic signals to mediate cellular response to stress, including activation of the p53 tumour suppressor pathway [81,89,90]. Under normal conditions, p53 is suppressed by HDM2 through either ubiquitin-mediated proteasomal degradation or by direct binding to p53, thereby inhibiting its transcriptional activity. However, in response to defects in Pol I transcription and ribosome biogenesis, several ribosome components and assembly factors, including rRNA and RPs, are released from the nucleolus into the nucleoplasm [81,89,90,91].

These factors, including RPL5/RPL11-5S rRNA complexes, interact with and inhibit HDM2, resulting in p53 stabilisation and activation of downstream responses such as DNA repair, cell cycle arrest, and apoptosis [90,91]. Nucleolar stress can also trigger p53-independent responses, which remain poorly understood [81]. Various cellular stresses, including DNA damage, serum starvation, depletion of nucleotides, and cell cycle arrest, inhibit Pol I inhibition and ribosome assembly, triggering the nucleolar stress response, highlighting that ribosome biogenesis is tightly coupled to cellular homeostasis and stress surveillance mechanisms [87,90,92,93].

## 4. Dysregulation of Ribosome Biogenesis and Function in Cancer

The notion that Pol I transcription and ribosome biogenesis are dysregulated in cancer had been postulated before the 1900s, based on the observation that the size and the number of nucleoli in malignant cells are greater than those in normal cells [1,2]. This association is further supported by studies of rare congenital disorders that affect ribosome function, collectively known as ribosomopathies. Individuals with ribosomopathies suffer from life-threatening conditions, including bone marrow failure and congenital malformations, and notably, have a higher risk of developing cancer in their lifetime [9,10]. Several known ribosomopathies can be classified into two main categories: those with mutations in RPs and those with mutations in other RBFs. Research groups have investigated how specific gene mutations in these ribosomopathies impact ribosome biogenesis, and the putative mechanisms behind cancer predisposition [94,95]. Additionally, recurrent somatic mutations in RPs and deregulation in RP expression have been identified in many types of cancer [94,95].

### 4.1. From Ribosomopathies to Cancer

Amongst all ribosomopathies, Diamond–Blackfan anaemia (DBA), a congenital red blood cell aplasia, is probably the best characterised. Roughly 70–80% of cases are ascribed to loss-of-function mutations in RP genes, which result in impairment of ribosomal maturation, a distinguishing feature of DBA [96,97]. Genetic lesions in 24 RP genes have been linked to DBA to date. This includes genes encoding RPs of the small 40S (RPS7, RPS10, RPS15A, RPS17, RPS19, RPS20, RPS24, RPS26, RPS27, RPS27A, RPS28, RPS29) and the large 60S (RPL5, RPL8, RPL9, RPL11, RPL15, RPL17, RPL18, RPL26, RPL27, RPL31, RPL35, RPL35A) ribosomal subunits [98]. Studies have shown that RP haploinsufficiency caused by mutations in these genes consequently depletes free 40S and 60S subunits, and thus functional 80S ribosomes [98,99]. Across DBA-related RP genes, RPS19 (25%), RPS26 (7%), RPL5 (7%), RPL11 (5%), and RPL35A (4%) were found to be more commonly altered [99]. More recently, defects in non-RP genes, GATA1, TSR2, and HEATR3, have also been associated with the DBA phenotype, though they account for only a small proportion of all cases [100,101,102]. About 80% of DBA cases initially respond to corticosteroid therapy [99,103]. Some individuals achieve spontaneous remission, defined as having an adequate haemoglobin level for at least 6 months without transfusion, by early adulthood [99]. A widely perceived mechanism through which corticosteroids stimulate red blood cell production in DBA is induction of stress erythropoiesis, involving an expansion of erythroid progenitors and the activation of glucocorticoid receptors [104,105,106]. Studies also indicate that corticosteroids contribute to the development of red blood cells by modulating signalling pathways, like p53, MYC, and mTOR [105,107]. However, the precise mechanism of corticosteroid response in DBA remains elusive due to the complexity of the disease and the multifaceted effects of corticosteroids [107]. Individuals with DBA are nearly five times more likely to develop solid tumours and haematological malignancies compared with the general population [94,108,109]. Different types of cancers with varying incidence rates have been reported in DBA patients, including gastrointestinal cancers, sarcomas, leukemia, myeloma, and lymphoma [108,109]. Notably, in DBA, there is an increased risk of myelodysplastic syndrome (MDS), which can progress into acute myeloid leukemia (AML).

Ribosomopathies can also be caused by mutations in non-RP genes necessary for ribosomal assembly and maturation. Shwachman–Diamond syndrome (SDS), dyskeratosis congenita (DC), cartilage–hair hypoplasia (CHH), and Treacher Collins syndrome (TCS) exemplify this group [9,94]. In SDS, mutations in the SBDS gene and related factors disrupt 60S subunit maturation, leading to reduced 80S ribosome formation and impaired protein synthesis [110]. X- linked DC is associated with mutations in DKC, affecting rRNA pseudouridylation, a modification step in ribosome biogenesis, whilst CHH arises from RMRP mutations, which hinder pre-rRNA cleavage [111,112]. TCS is linked to TCOF1 and POLR1 gene mutations that disrupt Pol I-driven rRNA transcription and modification [113]. Notably, ribosomopathies like SDS, DC, and CHH are all associated with a higher cancer risk, including MDS, AML, and various solid tumours, suggesting that defects in ribosome biogenesis contribute to oncogenic transformation [94,95,114]. This link may reflect selective pressure for malignant clones that adapted to chronic ribosomal stress; however not all ribosomopathies, such as TCS, exhibit this predisposition [94,95].

It is important to note that defects in ribosome biogenesis, whilst significant, are not on their own sufficient to drive cancer development. This can be inferred from studies of ribosomopathies, where individuals initially present with clinical phenotypes attributed to cellular hypo-proliferative defects, such as bone marrow failure, malformations, and mental and motor deficiencies [94]. Subsequently, cells acquire secondary mutations in growth signalling pathways to transform into hyper-proliferative states and become malignant. Sulima et al. proposed a model where defects in ribosome biogenesis create selective pressure on cells, leading to compensatory mutations that rescue them from hypo-proliferative states [94]. These rescued cells acquire malignant properties, such as altered proliferation capacity and protein synthesis, evading the control of ribosome production and resulting in clonal expansion.

### 4.2. Oncoribosomes

Whilst the causal link between ribosomal defects and cancer development remains unclear, dysregulation of ribosome biogenesis is implicated in cancer progression through the identification of recurrent somatic mutations in RP genes in several types of cancer [95]. A large-scale genomic study in various cancer samples and cell lines found that 43% harbour heterozygous deletions in RP genes [115]. Comparable to the germ-line mutations in RP genes observed in DBA, these somatic RP gene lesions result in haploinsufficiency of the corresponding RPs. Recurrent mutations in genes of the large 60S subunit components RPL5, RPL10, RPL11, RPL22, and RPL23A, and the small 40S subunit components RPS15, RPS20, and RPS27 have been specifically implicated in cancer [95,116,117,118,119].

Across RP genes, the frequency of mutations in or deletions of RPL5 is relatively high, with several reported cases in breast cancer, multiple myeloma (MM), melanoma, glioblastoma, and T-cell acute lymphoblastic leukemia (T-ALL) [95,116,120,121]. Further studies in MM indicate that patients with low RPL5 expression had unfavourable outcomes [117]. RPL5, along with RPL11, contributes to p53 stabilisation by inhibiting the E3 ubiquitin–protein ligase HDM2, thereby linking impaired nucleolar stress response to tumorigenesis [122]. Analysis of 19,000 cancer samples from The Cancer Genome Atlas (TCGA) and International Cancer Genome Consortium (ICGC) revealed mutations in RPL5 and RPL11 in 139 and 74 cases, respectively, indicating their involvement in cancer pathogenesis [122]. Mutations in RPL10 and RPL22 have also been implicated in tumorigenesis and identified in additional cancer types, including the rare somatic RPL10 mutations in MM and RPL22 mutations producing truncated proteins in endometrial, gastric and colorectal cancer samples [119]. Moreover, amplification of the genomic region containing the RPL23A gene has been uncovered in samples of uterine cancer [116].

Whilst mutations in 40S RP genes are less common, RPS15 is notable for being mutated or deleted in up to 20% of chronic lymphocytic leukemia (CLL) cases, impairing p53 activation [123,124]. Similarly, RPS20, which also interacts with HDM2, is mutated in colorectal cancer [125]. Furthermore, mutations in RPS27 have been detected in melanoma samples [126]. This evidence of recurrent somatic mutations in a variety of RP genes across different cancer types highlights the striking association of ribosome alterations in oncogenesis.

Beyond quantitative alterations in ribosome abundance, qualitative differences in RP composition, rRNA sequence, and rRNA modifications give rise to ribosome heterogeneity. Emerging evidence supports the concept that heterogeneous ribosomal subpopulations perform specialised functions in translational control, enabling context-dependent regulation of gene expression [127]. Specific ribosomes have been shown to preferentially translate mRNAs with complex secondary structures, distinct sequence motifs, or stress-responsive elements. Recent advancement in high-resolution imaging and single-particle profiling techniques now allow direct visualisation of individual ribosomes, facilitating the characterisation of their ribosome composition and spatial organisation [128].

In many tumour types, changes in ribosomal protein composition and rRNA modifications give rise to so-called oncoribosomes that support elevated protein synthesis and regulate the translation of tumour-specific proteins. For example, the RPL10-R98S mutation frequently found in T-ALL promotes the translation of mRNAs containing internal ribosome entry sites (IRES) in their 5′UTR, such as BCL2, thereby conferring survival advantage [129]. Additionally, this mutation impairs programmed −1 ribosomal frameshifting, a key mechanism of regulating gene expression, and stabilises mRNAs involved in the JAK-STAT signalling pathway [130]. Post-transcriptional modifications of rRNA also contribute to ribosome heterogeneity. 2′-O-methylation, catalysed by FBL, and pseudouridylation, mediated by DKC1, are key modulators. FBL overexpression, often observed following p53 loss, drives IRES-dependent translation of oncogenes such as IGF1R, MYC, and VEGFA, promoting tumorigenesis [131]. FBL also interacts with YBX1 to activate BRCA1 transcription, highlighting its extra-ribosomal function beyond ribosome biogenesis in promoting chemoresistance in cancer [132]. Conversely, mutations in DKC1 lead to reduced rRNA pseudouridylation, impairing the translation of tumour suppressors (p53, p27^Kip1^) and anti-apoptotic proteins (BCL-XL, XIAP) [133,134]. Additionally, oncoribosomes can reprogram cancer metabolism, as seen in T-ALL, where RPL10-R98S mutations induce oxidative stress and deregulated serine/glycine metabolism [135,136]. Together, these findings underscore the role of ribosome specialisation in promoting oncogenic translation programs and metabolic reprogramming. The concept of oncogenic ribosome thus represents both a driver of tumorigenesis and a promising target for selective cancer therapies.

### 4.3. Oncogenic Signalling Modulates Ribosome Synthesis and Function

Several oncogenic pathways converge to upregulate ribosome biogenesis and mRNA translation, creating a feed-forward network that fuels “translation addiction” in cancer cells. Cancer-associated genetic alterations in MYC, PIK3CA, PTEN, KRAS, and BRAF dysregulate growth-promoting signals, enhancing ribosome production and protein synthesis [137,138].

MYC is a master regulator in coordinating ribosome biogenesis at a transcriptional level through its involvement in initiating Pol I, II, and III actions (Figure 4). In the nucleolus, MYC promotes the assembly of a PIC at the rRNA gene promoters, driving the synthesis of 18S, 5.8S and 28S rRNAs. MYC also stimulates Pol II transcription of genes encoding RPs and RBFs (e.g., DKC1, UBF, FBL, NPM1 and NCL mRNAs) and enhances Pol III activity by interacting with Pol III-specific transcription factor IIIB (TFIIIB) [139]. The MYC-TFIIIB complex engages with TRRAP and GCN5, leading to Pol III transcription of the 5S rRNA and tRNA genes. In many cancers, however, MYC is frequently overexpressed and hyperactivated through gene translocations, amplifications or upstream oncogenic signalling stimulation, resulting in elevated levels of rRNAs, RPs, RBFs, and key translation initiation components such as the eIF4F complex and 5′ mRNA cap-methylation machinery [140,141]. This broad MYC-driven transcriptional program sustains high translational capacity and drives uncontrolled proliferation.

In addition to MYC, the mechanistic target of rapamycin (mTOR) complex 1 (mTORC1) pathway orchestrates ribosome biogenesis and mRNA translation in response to extracellular growth signals (Figure 4). Binding of growth factors, cytokines, and hormones to cell-surface receptor tyrosine kinases (RTKs) and G protein-coupled receptors (GPCRs) activates phosphatidylinositol 3-kinase (PI3K)/Akt signalling [142,143]. Activated Akt inhibits tuberous sclerosis complex 2 (TSC2), a GTPase activating protein (GAP) that keeps Ras homolog enriched in brain (RHEB) protein in its inactive, GDP-bound from. The relief of TSC2 inhibition allows GTP-bound RHEB to activate mTORC1, a kinase complex containing mTOR, Raptor, and mLST8. In cancers harbouring PIK3CA mutations or PTEN loss, PI3K/Akt-mTORC1 signalling is constitutively active [144]. Moreover, Ras–ERK signalling driven by mutant KRAS, HRAS, NRAS, or BRAF can activate mTORC1 independently of PI3K/Akt via ERK and p90 ribosomal S6 kinase (RSK)-mediated phosphorylation of TSC1 and mTORC1 component Raptor [145].

Once activated, mTORC1 phosphorylates its two main substrates, p70 RSK (referred to as S6K1/2) and eIF4E-binding proteins (4E-BPs) [146,147]. Phosphorylation of S6K1 stimulates translation initiation through RPS6 phosphorylation, eIF4B activation, and programmed cell death 4 (PDCD4) inhibition, thereby stimulating the mRNA helicase activity of eIF4A during translation initiation [148,149]. Additionally, mTORC1-mediated phosphorylation of 4E-BPs release their binding from cap-binding protein eIF4E, allowing the formation of the eIF4F complex and cap-dependent translation. S6K1 also promotes elongation by phosphorylating and inhibiting eEF2 kinase (eEF2K), thus increasing eEF2 activity [150]. Collectively, these mTORC1-driven signalling pathways promote global protein synthesis, underscoring its significance in supporting oncogenic growth. In addition to mTORC1, the Ras–ERK pathway regulates translation initiation through MAPK-interacting kinase 1 (MNK1) and 2 (MNK2) [151,152]. MNKs interact with the eIF4F complex via eIF4G, to phosphorylate eIF4E, increasing its cap-binding and enhancing recruitment of 40S subunit to oncogenic mRNAs. Collectively, oncogenic MYC, PI3K/Akt-mTORC1, and Ras–ERK/MNK signalling drive ribosome production and translational output to promote cancer cell proliferation, survival, and metastasis.

## 5. Targeting Ribosome Biogenesis and mRNA Translation in Cancer

Dysregulation in ribosome synthesis and the translation machinery are increasingly recognised as potential vulnerabilities that can be exploited in cancer therapy.

### 5.1. Existing Chemotherapeutics Affect Ribosome Synthesis

Chemotherapy has been established as a pillar of cancer treatment for decades. Whilst these agents are collectively recognised for their DNA damaging properties, research findings have revealed that some also impact ribosome biogenesis at various stages [153]. Classes of chemotherapeutic agents inhibiting the production of ribosomes include alkylating agents, DNA intercalating agents, antimetabolites, and topoisomerase inhibitors (Figure 5).

#### 5.1.1. Alkylating-like Platinum Compounds

Platinum-based compounds such as cisplatin and oxaliplatin have a similar mechanism of action to alkylating agents [154,155]. Whilst they do not harbour alkyl groups, these compounds primarily form covalent bonds between DNA nucleotides within the same strand, across opposite strands, and with DNA-associated proteins, leading to DNA-protein crosslinks [154]. These crosslinks alter the DNA structure, causing defects in DNA repair and replication. Beyond these effects on DNA, mechanistic studies have demonstrated that cisplatin-induced DNA crosslinks trigger a redistribution of rRNA transcription machinery components, including UBF, TBP, TAFs, and Pol I, ultimately inhibiting rRNA synthesis [156]. Further research confirmed that both cisplatin and oxaliplatin rapidly inhibited Pol I transcription of rRNA genes [153]. Notably, this inhibitory effect was observed at clinically relevant concentrations of oxaliplatin, whereas higher doses of cisplatin were required to achieve a similar outcome [157]. Additional comparisons between these two compounds revealed that cisplatin induces cell death through activation of the DDR, whilst oxaliplatin does so by triggering ribosome biogenesis stress leading to activation of the nucleolar stress response [158]. This suggests that selectively targeting ribosome function could serve as an alternative strategy for inducing cytotoxicity with minimal DNA damage.

#### 5.1.2. DNA Intercalating Agents

Key compounds that belong in this class of drugs include actinomycin D (ActD), mitomycin C (MMC), doxorubicin, and mitoxantrone. These molecules insert themselves between DNA base pairs, preventing replication and transcription, inhibiting topoisomerase (TOP) enzymes, and causing DNA damage [159]. ActD and MMC preferentially intercalate the DNA between guanine-cytosine (GC) base pairs, thereby interfering with transcription at GC-rich regions [160,161]. ActD also promotes the formation of transient TOP-DNA complexes. TOP enzymes exist in two forms, type I (TOP1) and II (TOP2), both of which create DNA breaks to relieve supercoiling tension during replication. Stabilisation of TOP-DNA structures for extended periods impedes the transcription machinery, inducing DNA strand breaks [162]. Beyond their DNA damaging effects, these compounds effectively inhibit rRNA transcription and 47S pre-rRNA production [153]. This is likely due to the presence of GC-rich regions at rRNA genes, where ActD and MMC predominantly act [33,90]. Notably, low doses of ActD (<5 nM) selectively inhibit Pol I transcription, whilst high doses of ActD inhibit Pol II and Pol III as well, leading to global transcription suppression [159].

#### 5.1.3. Antimetabolites

Antimetabolites are chemotherapeutic agents that resemble nucleotides but contain slight structural variations. They interfere with DNA and RNA synthesis through inhibiting essential enzymes and incorporating into DNA and RNA structures [163]. Pyrimidine analogue, 5-fluorouracil (5-FU), disrupts DNA replication by inhibiting thymidylate synthase, and impairs RNA synthesis by being inserted into various RNA species [164]. This interference affects post-transcriptional modifications of tRNA, snRNA, and rRNA, compromising their processing and stability [165]. Specifically, 5-FU interferes with the late processing and maturation steps of rRNA synthesis, leading to reduced 28S rRNA abundance and triggering a nucleolar stress response involving p53 stabilisation [166]. Another study has verified that 5-FU can be incorporated into rRNA of functional ribosomes, altering translation [167]. In addition, folate antagonist methotrexate that blocks purine and pyrimidine synthesis, also affect ribosome synthesis. Unlike 5-FU, methotrexate inhibits rRNA transcription as evidenced by decreased 47S pre-rRNA levels [153].

#### 5.1.4. Topoisomerase Inhibitors

TOP1 and TOP2 inhibitors represent an important class of chemotherapeutic agents. TOP1 inhibitors such as camptothecin (CPT) inhibit transcription and induce DNA damage by forming TOP1-DNA complexes [168,169]. Early evidence indicated the ability of CPT and its derivative, topotecan, to inhibit global RNA synthesis [170]. Another finding confirmed that this included rRNA synthesis suppression [34]. Interestingly, CPT has a stronger effect on early rRNA processing than on rRNA transcription [153]. Similarly, TOP2 inhibitors, particularly TOP2 poisons, impact multiple stages of ribosome biogenesis [153]. As mentioned previously, doxorubicin and mitoxantrone rapidly inhibit rRNA transcription. Another type of TOP2 poison, etoposide, which stabilises TOP2-DNA covalent complexes without intercalating the DNA, also affects ribosome biogenesis, specifically impairing rRNA processing [153,171].

These findings highlight that targeting the ribosome is one of the diverse mechanisms of action of chemotherapeutic agents. Therefore, developing novel compounds that selectively interfere with ribosome biogenesis could achieve greater therapeutic efficacy whilst also overcoming major limitations of chemotherapy, including severe side effects and secondary cancers.

### 5.2. Pol I Transcription Inhibitors

Efforts to develop highly specific inhibitors of Pol I transcription, the rate-limiting step of ribosome biogenesis, led to the development of compounds like CX-5461, BMH21, and PMR-116 (Figure 5).

High-throughput quantitative reverse transcription polymerase chain reaction (qRT-PCR)-based screening, measuring drug-related effects on transcription, led to the discovery of CX-5461, a small molecule inhibitor with greater selectivity towards inhibiting Pol I than Pol II [172]. CX-5461 blocks Pol I binding to the SL1 complex and its recruitment to rRNA gene promoters [172]. As a potent rRNA synthesis inhibitor, CX-5461 induces nucleolar stress, resulting in the activation of the p53 pathway [173]. Additionally, CX-5461 triggers other molecular responses, including activation of the DNA damage response (DDR), stabilisation of G-quadruplex (G4) structures, and TOP2 trapping leading to replication stress [174,175,176,177,178]. The mechanism by which CX-5461 traps TOP2 on the DNA differs to classical TOP2 poisons due to its selectivity towards genomic regions characterised by high transcriptional activity and enrichment of G4 structures [176,179]. These responses ultimately lead to cell cycle arrest and cell death.

Several genetic screens have uncovered biomarkers of response to CX-5461. Components of the homologous recombination (HR) repair machinery, the Fanconi anaemia pathway, and the G4-resolving polymerase POLQ were identified to be synthetic lethal with CX-5461 [180,181]. Various studies have also shown that CX-5461 sensitivity correlates with a high proportion of transcriptionally active rRNA gene repeats, BRCA mutations, and MYC activation, underscoring the importance of biomarker-guided patient selection for optimal therapeutic benefit [174,175,182,183,184]. CX-5461 has demonstrated significant efficacy in combination with various agents, enhancing its anti-tumour activity in preclinical cancer models. The combination of CX-5461 and low-dose topotecan markedly increased nucleolar replication stress and DDR, resulting in significant inhibition of high-grade serous ovarian cancer tumour growth [180]. CX-5461 was also shown to synergise with poly (ADP-ribose) polymerase inhibitors, providing improved therapeutic benefit in ovarian and prostate PDX models [175,185]. Furthermore, CX-5461 works in combination with PIM kinase, mTOR (everolimus), and histone deacetylase (panobinostat) inhibitors in MYC-driven cancer models, through amplifying the inhibition of ribosome biogenesis, protein synthesis, and metabolic homeostasis [186,187,188]. Resistance to CX-5461 has been attributed to the loss of TOP2 isoforms, TOP2A and TOP2B, which are essential for CX-5461-induced replication stress [176,189]. Other resistance mechanisms include translational rewiring that enhanced mitochondrial metabolism and activated pro-survival pathways [187].

Early clinical studies of CX-5461 provided proof-of-concept evidence supporting the therapeutic potential of targeting ribosome biogenesis [190,191]. In a first-in-human phase I study, CX-5461 demonstrated clinical benefit in 6/16 patients with advanced haematological malignancies [190]. This included a prolonged partial response in one patient with anaplastic large cell lymphoma and stable disease in five patients with DLBCL and MM [190]. That study demonstrated on-target effect of CX-5461 in reducing the rate of rRNA gene transcription in peripheral blood mononuclear cells and patient tumours [190]. A phase I/II trial of CX-5461 in heavily pre-treated patients with advanced solid tumours showed an overall response rate (ORR) of 14%, mainly in patients with HR-deficient cancers [191]. This validated preclinical findings of CX-5461 exhibiting synthetic lethality with DNA repair defects [175,178] and led to further investigations in a phase Ib expansion study (NCT04890613) of CX-5461 in patients with solid tumours and BRCA1/2, PALB2, and/or other HR mutations. Preliminary data reported 40% of 15 patients evaluable for response had clinical benefit [192]. Amongst these, five patients with BRCA1 (*n* = 4) and HR (*n* = 1) mutated ovarian cancer achieved stable disease, two of whom had prolonged response of more than 6 months [192]. All five ovarian cancer patients who achieved stable disease in this cohort had previously received platinum-based and PARP inhibitor therapy, indicating that CX-5461 has a different sensitivity profile to these compounds [175]. CX-5461 exhibited therapeutic potential through its multifaceted mechanisms of action. However, these mechanisms may also contribute to reported adverse effects, including photosensitivity and palmar-plantar erythrodysesthesia [190,191]. Notably, whilst ribosomopathies are associated with bone marrow failure syndromes, the inhibition of Pol I transcription by CX-5461 did not exhibit evidence of myelosuppression in the clinical studies [190,191]. Overall, these clinical trials demonstrated relatively low ORR; therefore, further evaluation of combination therapies is critical, such as the REPAIR study (NCT05425862) in prostate cancer which assesses the combination of CX-5461 and the PARP inhibitor, talazoparib. A recent study by Koh et al. using an immortalised cell line model in vitro suggests that CX-5461 can cause rapid, widespread, and non-selective mutagenesis [193]. Notably, several early-phase studies of CX-5461 in patients with HR-deficient and metastatic solid tumours are recruiting patients (NCT04890613, NCT06606990). These ongoing clinical efforts underscore the therapeutic promise of CX-5461.

A second-generation compound, PMR-116, was developed to provide a more selective inhibition of Pol I transcription, whilst eliminating the TOP2 poisoning activity and phototoxicity associated with CX-5461 [194]. PMR-116 exhibits improved pharmacokinetics, including enhanced tumour penetration and a longer plasma half-life, leading to better tumour suppression in preclinical models [194]. PMR-116 is being evaluated in a phase I clinical trial for patients with advanced solid tumours (CTRN12620001146987), aiming to assess its clinical efficacy and safety profile [194].

Another Pol I inhibitor, BMH-21, was identified through chemical compound library screening for p53-activating agents [195]. BMH-21 inhibits rRNA synthesis by intercalating onto GC-rich regions of rRNA genes, leading to degradation of the Pol I catalytic subunit RPA194 and activation of p53 without triggering a DDR [196]. Further studies revealed that BMH-21 inhibits Pol I transcription particularly at the elongation phase [197]. Preclinical models have also demonstrated its therapeutic potential across several types of cancer, though clinical testing has yet to be conducted [198].

In summary, the development of selective Pol I transcription inhibitors aims to exploit the reliance of cancer cells on dysregulated ribosome biogenesis, whilst minimising toxicity to normal cells. CX-5461, the first-in-class inhibitor of Pol I transcription, demonstrated promising therapeutic activity with a favourable safety profile. It has since been recognised as a multi-modal agent, also functioning as a TOP2 poison and a G4 stabiliser. These additional mechanisms of action contribute to its anti-tumour efficacy particularly through its synthetic lethal interactions with HR deficient cancers. The second-generation compound PMR-116 was developed to retain on-target inhibition of Pol I with improved pharmacokinetic properties, but the clinical efficacy and tolerability are not established yet. Similarly, BMH-21 has not yet progressed to clinical evaluation. Therefore, whilst selective Pol I transcription inhibitors present a novel class of cancer therapeutics, their clinical development is still in its early stages. Overcoming limitations, such as the incomplete understanding of their mechanisms of action, the absence of robust predictive biomarkers, and the need for rational combination strategies, will be essential to fully harness the therapeutic potential of selective Pol I inhibitors in cancer.

### 5.3. Direct Inhibitors of the Translation Machinery

In addition to impairing ribosome biogenesis, several strategies have been explored to directly interfere with the translation machinery, particularly through targeting components of the eIF4F complex, such as eIF4E, eIF4G, and eIF4A [199] (Figure 5).

#### 5.3.1. Inhibitors of eIF4E-eIF4G Interaction

The assembly of the eIF4F complex can be targeted by disrupting the interaction between cap-binding subunit eIF4E and scaffolding protein eIF4G. High-throughput screens have identified small molecule compounds, 4EGI-1 and 4E1RCat, which impede eIF4E-eIF4G interaction. 4EGI-1 displaces eIF4G from eIF4E, whilst stabilising the binding of 4E-BPs to eIF4E, thereby inhibiting cap-dependent translation [200]. Structural studies proposed an allosteric effect of 4EGI-1, where it binds eIF4E at a distant region to the eIF4G binding site and causes conformational changes preventing eIF4G from interacting with eIF4E [201,202]. Additional studies have shown that 4EGI-1 induces cancer cell death through the induction of pro-apoptotic protein NOXA and the repression of pro-survival factors BCL2A1 and BCL2L1 [203,204]. Furthermore, 4EGI-1 demonstrated synergy with the BCL2 inhibitor ABT-737 to enhance apoptosis in CLL and improve sensitivity to gemcitabine in non-small cell lung cancer (NSCLC) in vitro [205]. The subsequent generation BCL2 inhibitor, venetoclax (ABT-199), is presently approved for use in CLL and AML. Meanwhile, 4E1RCat interferes with eIF4F complex assembly by binding eIF4E at sites typically bound by eIF4G and 4E-BPs [206]. Cencic et al. showed 4E1RCat reduced eIF4E-dependent translation of mRNAs encoding MCL-1 and MYC in vitro and reversed chemoresistance in a MYC-driven lymphoma mouse model [206]. Additional studies revealed that 4E1RCat in combination with selective inhibitor of eIF2α dephosphorylation salubrinal decreased cancer cell viability and inhibited xenograft tumour growth [207]. However, these small molecule inhibitors of eIF4E-eIF4G interaction exhibit poor potency and limited biological impact at concentrations practical for therapeutic applications.

More recently, EGPI-1, another small molecule blocking the eIF4E-eIF4G interaction, was identified and shown to exhibit anti-tumour activity in vitro and in vivo [208]. EGPI-1 reduced phosphorylation levels of eIF4E and 4E-BP and expression of oncogenic proteins cyclin D1 and MYC in lung cancer cells [209]. In xenograft models, tumour growth was inhibited following treatment with EGPI-1. These findings indicate the potential of targeting eIF4E-eIF4G interaction, though further optimisation studies are needed to improve therapeutic efficacy.

#### 5.3.2. Inhibitors of eIF4E Cap-Binding Activity

Another approach to target eIF4E in the eIF4F complex is by blocking its association with capped mRNAs. The effects of cap analogues, molecules chemically modified to resemble the 5′ m^7^G cap structure on mRNAs, have been studied in in vitro models of cancer [210,211]. Synthetic cap analogues effectively reduce cap-dependent translation by binding to eIF4E and depleting the eIF4E pool accessible to mRNAs. However, these cap analogues exhibit poor permeability and stability in vivo. To overcome the inefficient delivery of cap analogues, Ghosh et al. developed 4Ei-1, a prodrug metabolised into cap analogue 7-benzyl guanosine monophosphate (7Bn-GMP) after cell uptake [212]; 7Bn-GMP was reported as a potent antagonist of eIF4E-cap interaction with high binding affinity to eIF4E compared to the m^7^G cap structure on mRNA [213,214], and 4Ei-1 was rapidly converted into 7Bn-GMP in cells and resulted in dose-dependent inhibition of cap-dependent translation in zebrafish embryos [212]. Notably, eIF4E expression level was reduced in response to 4Ei-1 and restored by proteasome inhibitors [215]. This indicates that 4Ei-1 inhibits cap-dependent translation initiation by interfering with eIF4E cap binding and promoting proteasomal degradation of eIF4E. Further studies have demonstrated that 4Ei-1 decreases cell viability in vitro and enhances gemcitabine cytotoxicity in breast and lung cancer cells [215]. More recently, an acyclic nucleoside phosphonate prodrug strategy was utilised to create cap analogue prodrugs containing a bis-pivaloxymethyl (POM) (Z)-4-phosphono-but-2-enyl linker [216]. Bis-POM is a lipophilic portion of the structure masking the negative charge of the phosphonate group, allowing increased cell permeability. The top performing compound was shown to decrease expression of oncogenic proteins whose translation was mediated by eIF4E, displaying anti-proliferative activity in vemurafenib-resistant melanoma cells [217]. Whilst these cap analogue prodrugs can inhibit cap-dependent translation, they exhibit low potency and are unsuitable for further advancement as drug candidates.

Ribavirin, a currently available antiviral medication, was reported to directly bind eIF4E and inhibit eIF4E-cap interaction, though several studies have disputed its mechanism of action [199,218,219]. Indeed, ribavirin has demonstrated anti-tumour activity in preclinical studies and therapeutic benefit in clinical trials, but its efficacy may involve mechanisms unrelated to eIF4E [220,221,222,223].

#### 5.3.3. Antisense Oligonucleotides Targeting eIF4E

Targeting eIF4E can also be achieved through antisense oligonucleotide (ASO)-based strategies [224]. ASOs are short, single-strand oligonucleotides designed to bind specific RNA sequences through complementary base pairing and induce RNase H-mediated degradation of target mRNA. Early preclinical findings indicated that ASO LY2275796 (also referred to as ISIS 183750) achieved on-target silencing of eIF4E mRNA in human tumour xenograft models, significantly suppressing tumour growth [224]. In a phase I dose escalation study, a reduction in eIF4E expression level was detected in tumour biopsies following LY2275796, although no tumour response was observed [225]. LY2275796 was evaluated in combination with TOP1 inhibitor irinotecan in a phase I/II study (NCT00903708) for the treatment of irinotecan-refractory colorectal cancer [226]. Whilst no objective responses were reported, 7/15 patients displayed stable disease [226]. No new clinical trials of LY2275796 have been registered.

#### 5.3.4. Targeting RNA Helicase eIF4A

The prominence of the DEAD box RNA helicase eIF4A in mRNA translation has also led to the evaluation of natural compounds with the ability to inhibit its activity, such as hippuristanol, silvestrol, and pateamine A (PatA). Hippuristanol directly binds to the C-terminal domain of eIF4A, exerting an allosteric effect that blocks RNA binding and eIF4A helicase activity [227]. This results in the inhibition of cap-dependent translation initiation and formation of 80S ribosomes. Mechanistic studies indicate that hippuristanol induces cell cycle arrest and apoptosis associated with decreased cyclin D1, CDK4, and CDK6, and the activation of caspases in vitro and in vivo [228]. Hippuristanol has also been shown to enhance lymphoma and leukemia cell death in combination with BCL2 inhibitor ABT-737 [229]. Silvestrol belongs to a class of eIF4A inhibitors called rocaglates that harbour a bicyclic cyclopenta-[b]benzofuran core [230,231]. Unlike hippuristanol, these compounds stabilise the interaction between eIF4A and polypurines within the 5′ UTR, preventing eIF4A function in the eIF4F complex and creating steric barriers that block ribosome recruitment [232]. Rocaglates, including silvestrol, have been shown to trigger anti-tumour activity associated with induced apoptosis and inhibit the translation of oncogenic mRNAs in preclinical models [230,233,234]. Further studies have also demonstrated that silvestrol sensitises tumours with PTEN inactivation and elevated eIF4E to doxorubicin in vivo [235]. Similar to silvestrol, PatA stimulates the dimerization of eIF4A and non-specific RNAs, inhibiting eIF4A function and translation initiation [236,237]. Both PatA and the synthetic derivative des-methyl des-amino PatA (DMDAPatA) were shown to be potent inhibitors of translation initiation, harbouring anti-proliferative activity in vitro [238,239]. Despite their promise, there are many challenges in the clinical development of these eIF4A inhibitors due to toxicity concerns and the chemical complexity of natural products. Silverstrol exhibits low oral bioavailability and is a substrate of the multidrug efflux pump which plays a significant role in drug resistance [240,241]. In addition, PatA binds eIF4A irreversibly, causing persistent inhibition of protein synthesis and increased toxicity in vivo [238]. DMDAPatA was initially developed to overcome these issues, though its potency is limited due to high plasma protein binding [242].

More recently, Ernst et al. successfully optimised candidate drug eFT226 (Zotatifin) based on the structure of rocaglate derivative rocaglamide A [243]. eFT226 demonstrated highly selective eIF4A-mediated translation inhibition, which resulted in the downregulation of eIF4A-dependent oncogenic proteins [243,244]. The compound also exerted anti-tumour activity, indicated by the induction of cell cycle arrest and apoptosis breast cancer cell line MDA-MB-231 [245]. In orthotropic xenograft models, eFT226 suppressed tumour growth without significant toxicities. Additional studies revealed that eFT226 exerts anti-proliferative effects in a panel of solid tumour cell lines with HER2 and FGFR1/2 RTK dysregulation [245]. It was also shown that RTK-driven tumour cells with elevated mTOR signalling and increased basal rates of nascent protein synthesis were more sensitive to eFT226. Furthermore, eFT226 synergises with PIK3CA inhibitor alpelisib and Akt inhibitor ipatasertib to induce cytotoxicity in RTK-driven tumours in vitro and in vivo. Clinical studies to examine eFT226 in advanced solid tumours (NCT04092673) and breast cancer (NCT 05101564) are ongoing.

#### 5.3.5. Translation Elongation Inhibitor: Omacetaxine

Another compound, omacetaxine (also known as homoharringtonine), inhibits the translation machinery at the elongation stage. Unlike the previously mentioned therapeutic agents which target translation factors, omacetaxine directly binds to the A-site cleft in the 60S ribosome subunit [246]. This unique mechanism prevents aminoacyl-tRNAs from interacting with the ribosome, blocking subsequent peptide bond formation and translation elongation [246]. Omacetaxine has been approved by the FDA for the treatment of chromic myeloid leukemia (CML) [247,248]. A more recent study in relapsed/refractory MM demonstrated that omacetaxine synergises with immunomodulatory drugs, causing substantial downregulation of the oncoproteins interferon regulatory factor 4 (IRF4) and MYC [249].

Altogether, these findings highlight the potential of targeting the translation machinery in cancer therapy. As cancer cells frequently exhibit “translation addiction” and rely on hyperactivated cap-dependent translation to sustain growth and survival, targeting components of the eIF4F complex presents an attractive approach. Various strategies have been explored, including small molecules disrupting eIF4F complex assembly (e.g., 4EGI-1, 4E1RCat), ASOs (e.g., LY2275796), cap analogues (e.g., 4Ei-1), and inhibitors of RNA helicase eIF4A (e.g., hippuristanol, silvestrol). Whilst several agents have demonstrated promising anti-tumour activity in preclinical models and early-phase trials, clinical translation has been limited by issues relating to suboptimal pharmacokinetics, off-target toxicity, poor selectivity, and the lack of predictive biomarkers. Despite these challenges, ongoing efforts to optimise drug delivery, develop more selective analogues (e.g., eFT226), and identify biomarkers of “translation addiction” continue to drive the development of translation-targeting therapies.

### 5.4. Inhibitors of Signalling Pathways Regulating Translation

In addition to direct inhibitors of the translation machinery, mRNA translation can also be suppressed through targeting regulatory signalling pathways, such as the PI3K/ Akt/mTOR and MAPK/ERK pathways (Figure 5).

#### 5.4.1. mTOR Inhibitors

The first generation of mTOR inhibitors, rapamycin and the rapalogues everolimus and temsirolimus, inhibit mTORC1 in the PI3K/Akt/mTOR pathway [250]. They form a complex with FK506-binding protein FKBP12 that binds to the FRB domain of mTOR, causing allosteric inhibition of mTORC1 [251]. This limits the phosphorylation of downstream target S6K1, and to a lesser extent 4E-BPs [251]. Studies have demonstrated the anti-tumour activity of these compounds in various preclinical cancer models, with everolimus and temsirolimus approved by the FDA for the treatment of largely advanced-staged cancers [252]. However, their clinical efficacy remains limited, due to incomplete inhibition of 4E-BPs and compensatory mTORC2-induced activation of Akt, promoting cell survival and resistance [253,254,255].

To tackle these problems, ATP-competitive mTOR inhibitors and dual PI3K-mTOR inhibitors, were developed [256,257,258,259,260,261]. Several ATP-competitive mTOR inhibitors, including OSI-027, MLN-0128 (also known as Sapanisertib), and AZD2014, have been evaluated in clinical studies [262,263,264]. These compounds bind to the ATP-binding pocket of mTOR KIN domain, resulting in the inhibition of both mTORC1 and mTORC2. Preclinical findings show that these compounds are highly selective towards mTOR as opposed to other kinases such as P13K, significantly inhibiting S6K and 4E-BP phosphorylation and inducing anti-tumour activity [262,265]. However, the single-agent efficacy of these inhibitors in early clinical trials of advanced solid tumours and acute lymphoblastic leukemia (ALL) remain limited [266,267]. Notably, RapaLink-1, an mTOR inhibitor that targets both FRB domain and ATP-binding site of mTOR, was designed to overcome limitations of previous mTOR inhibitors [268]. Although RapaLink-1 has demonstrated growth-inhibitory effects in vitro and in vivo, it has yet to be evaluated in clinical settings. Dual PI3K-mTOR inhibitors were developed primarily to efficiently inhibit Akt, in addition to the downstream targets of mTOR. Like mTOR inhibitors, the dual PI3K-mTOR inhibitors have shown promising anti-tumour activity in various preclinical models and tested in clinical trials of advanced solid tumours and non-Hodgkin lymphoma [269,270,271]. Despite their improved potency, both ATP-competitive mTOR inhibitors and dual PI3K-mTOR inhibitors fail to fully overcome the resistance mechanisms of classical mTOR inhibitors [199,272], underscoring the complexity of targeting mTOR signalling in cancer therapy.

#### 5.4.2. MNK Inhibitors

Translation inhibition can also be achieved by targeting MNK1/2 in the MAPK/ERK signalling pathway. Like mTOR inhibitors, selective MNK1/2 inhibitors have been heavily investigated over the past few decades [273]. High-throughput screening, in which numerous compounds were tested for their selectivity towards MNK1/2 compared to other kinases, were used [274,275]. Most small molecule inhibitors of MNK1/2 are ATP-competitive inhibitors that bind to the active conformation of MNK1/2 at its ATP binding pocket [276,277]. These inhibitors, including initial MNK 1/2 inhibitor cercosporamide, were found to effectively block eIF4E phosphorylation and induce anti-tumour activity in various preclinical cancer models [274,276]. Amongst them, three MNK1/2 inhibitors, BAY1143269, eFT508, and ETC-206, have been further evaluated in clinical trials.

BAY1143269, which features a characteristic amino-substituted imidazo [1, 2-b] pyridazine structure, binds and inhibits MNK1 and MNK2 at half-maximal inhibitory concentration (IC_50_) of 40 nM and 904 nM [275]. It was shown to inhibit eIF4E phosphorylation, exhibiting anti-tumour properties both in vitro and in vivo [275,278]. Notably, response to BAY1143269 was associated with deregulation of the pro-survival protein survivin, leading to the induction of cell cycle arrest and apoptosis [275]. When BAY1143269 was combined with docetaxel, a therapeutic agent known to induce survivin, tumour growth in xenograft models of NSCLC was suppressed [275]. The safety and efficacy of BAY1143269 was assessed in a phase I study (NCT02439346), though the trial was terminated prematurely and no outcomes were published.

Similarly, eFT508 is a highly selective and potent inhibitor of MNK1/2, with IC_50_ values of 2.4 nM and 1.0 nM against MNK1 and MNK2, respectively [279]. It contains a key pyridone-aminal structure designed to fit within the ATP-binding pocket of MNK1/2 and form hydrogen bonds stabilising its interaction [279]. Preclinical findings indicate that eFT508 effectively blocks eIF4E phosphorylation and results in tumour growth inhibition in various cancer models [279]. These promising results have led to further clinical assessments into the safety and efficacy of eFT508 in advanced solid tumours, including prostate, colorectal, and breast cancers [280,281]. A recent publication reported that eFT508 effectively inhibits MNK1/2 activity in metastatic breast cancer tissues and can be safely combined with paclitaxel for subsequent phase II studies [281]. Notably, Xu et al. demonstrated that eFT508 significantly reduced the abundance of programmed death-ligand 1 (PD-L1), a key immune checkpoint protein often elevated upregulated in cancer cells to evade immune evasion surveillance [282]. Their study proposed that tumours driven by MYC and KRAS oncogenes possess altered mRNA translation, which makes them susceptible to eFT508. By suppressing PD-L1 translation, eFT508 reversed the metastatic nature of these tumours. This finding provided the rationale for combining eFT508 with anti-PD-L1 checkpoint inhibitors in clinical trials [282]. Preliminary findings of the phase II trial assessing the addition of eFT508 to checkpoint inhibitor therapy indicated that some patients with NSCLC, gastric, and renal cancer experience objective responses [283].

The third compound, ETC-206, also displays selectivity for both MNK1 (IC50 = 62 nM) and MNK2 (IC50 = 86 nM) and prevents phosphorylation of downstream target eIF4E [277]. It harbours an imidazo [1, 2-a] pyridine core structure that similarly occupies the ATP-binding site of MNK1/2 [277]. In addition to its anti-tumour activity as a single agent, studies show that ETC-206 synergizes with the BCR-ABL1 kinase inhibitor dasatinib, currently approved for the treatment of CML, to enhance tumour growth suppression in vivo [277]. Furthermore, recent findings indicate that ETC-206 was able to reverse multidrug resistance mediated by efflux transporter protein ABCG2 [284]. ETC-206 is currently being evaluated in combination with pembrolizumab or irinotecan for metastatic colorectal cancer in a phase II trial [285]. Interim data from the study found that ETC-206 was well-tolerated, prolonging the median time to progression and overall survival [285].

Despite the promise of MNK inhibitors, resistance mechanisms like those observed in mTOR/PI3K pathway inhibitors may arise [273]. Reactivation of MAPK/ERK signalling or other compensatory pathways could diminish their effectiveness, highlighting the need for combination therapies to sustain therapeutic responses.

### 5.5. Nuclear Export Inhibitor: Selinexor (KPT-330)

Over the past decade, selective inhibitors of nuclear export (SINE) have emerged as promising therapeutics by targeting nuclear-cytoplasmic transport of macromolecules (Figure 5). XPO1 is a well-characterised nuclear export receptor responsible for transporting about 220 proteins bearing a nuclear export signal (NES) [286,287]. Amongst the proteins are tumour suppressors (such as Rb, p53, APC, FOXO), immune response mediators (such as the inhibitor of NF-κB, IκB) and cell cycle regulators (such as p21, cyclin B1/D1) [288,289,290,291,292,293]. XPO1 is also involved in transporting of a subset of mRNAs associated with the cap-binding protein eIF4E, particularly those encoding proteins involved in cell survival, proliferation, metastasis, and invasion (such as MYC, cyclin D1, and HDM2) [294,295]. In addition, XPO1 is also responsible for translocating pre-40S and -60S ribosomal subunits composed of rRNA from the nucleus to the cytoplasm [58]. Proteomic studies further suggest that XPO1 regulates ribosome maturation, translation, and mRNA degradation [296].

The critical biological role of XPO1 and its overexpression in various solid and haematologic malignancies have driven the development of SINE compounds as a targeted cancer therapy [297,298,299]. These small molecule inhibitors covalently bind to residue Cys^528^ in the NES binding site of XPO1 in a slowly reversible manner, abrogating its nuclear export activity [300]. This causes nuclear retention and activation of tumour suppressor proteins and immune response mediators, whilst limiting nuclear-cytoplasmic export and translation of eIF4E-bound mRNAs [287,291,293]. Some of these mRNAs encode for cell cycle promoters, like cyclin D1/E and cyclin-dependent kinases, and anti-apoptotic proteins, like MCL-1 and BCL-xL [291,293]. As a result, growth signalling pathways such as Akt/PTEN/mTOR (via FOXO) and Wnt/β-catenin (via APC) are downregulated, leading to cell cycle arrest and apoptosis. Whilst the impact of XPO1 inhibition on ribosome biogenesis is less documented, studies using leptomycin B, a prototypical nuclear export inhibitor, have shown that blocking XPO1 reduces the export of pre-40S and -60S ribosomal subunits into the cytoplasm [58]. This suggests that disrupting ribosome synthesis and mRNA translation activity may be a critical mechanism through which SINE compounds exert their anti-cancer effects.

Across the SINE compounds developed, Selinexor (KPT-330) exhibited potent therapeutic efficacy with superior pharmacokinetic properties in various preclinical cancer models [287,301]. These findings supported the clinical investigations of Selinexor in haematologic malignancies and solid tumours [287,302]. Several clinical trials have evaluated the efficacy of Selinexor as a single agent and in combination with the standard-of-care therapeutics in heavily pre-treated MM [303,304,305]. The phase II STORM trial reported 26% of patients with triple-class refractory MM having partial response to Selinexor plus dexamethasone, with two patients achieving a complete response [303]. In the phase III BOSTON trial, the combination of Selinexor, bortezomib, and dexamethasone resulted in a median progression-free-survival (PFS) of 13.9 months compared with 9.5 months in the bortezomib–dexamethasone group [304]. The multi-arm STOMP study assessing the combination of Selinexor and backbone treatments for MM also reported ORR of 73% in the daratumumab group, 78% in the carfilzomib group, and 40% in the pomalidomide group [305,306,307,308]. These positive results led to FDA approval of Selinexor in combination with bortezomib–dexamethasone for MM patients who have received at least one prior line of therapy.

Selinexor has also shown promising activity in other haematological cancers, including AML, CLL and non-Hodgkin lymphoma [309,310,311,312]. The keystone phase II SADAL trial which reported an ORR of 28% (36/127) in patients with DLBCL, led to FDA approval of Selinexor for the treatment of relapsed/refractory DLBCL [312,313]. Clinical trials have also tested Selinexor in solid cancers, including glioblastoma, soft tissue and bone sarcoma, dedifferentiated liposarcoma, ovarian, cervical, and endometrial cancers [314]. The response rate across these malignancies varied, though the therapeutic potential of Selinexor remains evident [287].

Whilst Selinexor has demonstrated clinical efficacy, several clinical challenges limit its broader use and long-term benefits. Adverse events including gastrointestinal toxicity, fatigue, and weight loss are significant, frequently leading to dose interruption or discontinuation [315]. Patients also face common haematologic and metabolic complications, such as thrombocytopenia, neutropenia, and hyponatremia. These dose-limiting toxicities can be attenuated through lower doses and supportive care with antiemetics and growth factor support [316]. Furthermore, the mechanisms of acquired resistance to Selinexor are still under investigation. Potential factors contributing to resistance may involve compensatory upregulation of nuclear transport of FOXO3a through activation of the Akt pathway, activation of the NRG1-ERBB3 pathway, downregulation of TGF β-SMAD4 activity, or altered XPO1 binding via XPO1 Cys528 mutation [317,318,319]. Notably, through analysis of MM cells derived from patients enrolled in the BOSTON study, Restrepo et al. reported a three-gene signature, ETV7, WNT10A, and DUSP1, that predicts response to Selinexor-based therapy [320]. The study suggests that upregulation of these genes increases interferon-mediated apoptotic signalling, priming tumour cells to Selinexor therapy. Another study proposed XPO1, NF-kB (p65), MCL-1 and p53 protein levels as potential markers of response to XPO1 inhibitors in haematological malignancies [321].

Beyond its direct effect on tumour cells, Selinexor has been reported to exhibit immunomodulatory effects, which opens new opportunities for new combinations to enhance the efficacy and durability of response to immunotherapy. Preclinical studies have reported that Selinexor modulates immune cell functions through polarisation of macrophage cells to the anti-tumorigenic, immunostimulatory M1-like phenotype, improvement of NK cell function, and modulation of T-cell checkpoints [322,323,324]. Early clinical trial data assessing Selinexor as a bridging therapy for CAR-T therapy showed promising results, where patients with relapsed/refractory non-Hodgkin lymphoma and MM with patients achieving durable responses [325,326]. Optimising dosing strategies, developing biomarkers for improved patient selection, and developing better-tolerated second-generation XPO1 inhibitors (e.g., eltanexor) are active areas of investigation to enhance its therapeutic advancement.

## 6. Conclusions

Increased ribosome synthesis and altered mRNA translation are hallmarks of cancer cells that increase the anabolic demands required for malignant transformation and tumour growth. Targeting ribosome biogenesis and mRNA translation is a relatively new paradigm for treating cancer, showing encouraging preclinical and early-phase clinical activity. However, the functional complexity and heterogeneity of ribosomes in different cellular contexts complicate the development of selective inhibitors. Disruption of protein synthesis through a global suppression approach may prove effective in oncogene-driven cancers reliant on hyperactivated ribosome biogenesis. Alternatively, preferential inhibition of specific mechanisms that promote oncogenic mRNAs translation may have promising application in distinct genetic and molecular contexts, such as RP-mutated cancers. Equally as important is the development of predictive biomarkers for patient stratification to facilitate successful clinical translation of ribosome-targeting therapies.

Nevertheless, like other cancer therapeutic strategies, both intrinsic and acquired resistance pose a major challenge. Potential mechanisms associated with resistance to this new therapy can include alterations in ribosome composition, activation of compensatory signalling pathways, or adaptive translational reprogramming. Therefore, advancing the development of selective inhibitors, deepening our understanding of ribosome heterogeneity and translational addiction in cancer, and exploring effective combination strategies will be key to enhancing the clinical potential of ribosome-targeting therapies.

## Figures and Tables

**Figure 1 cancers-17-02534-f001:**
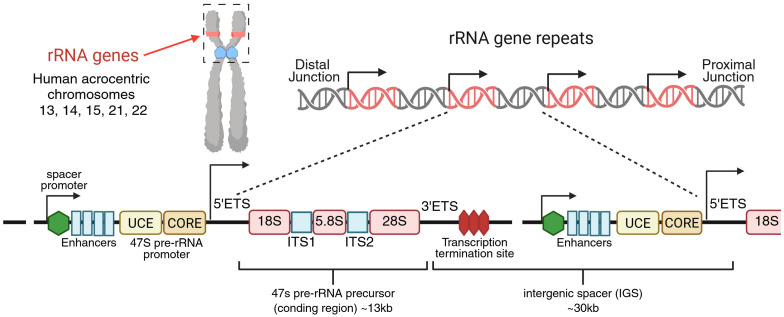
A canonical rRNA gene unit contains the 47S pre-rRNA coding region, embedded with 18S, 5.8S, and 28S rRNA sequences separated by ETS and ITS, and the intergenic spacer (IGS). The IGS consists of the core promoter and UCE elements, enhancers, and transcription termination site. (Created in https://BioRender.com).

**Figure 2 cancers-17-02534-f002:**
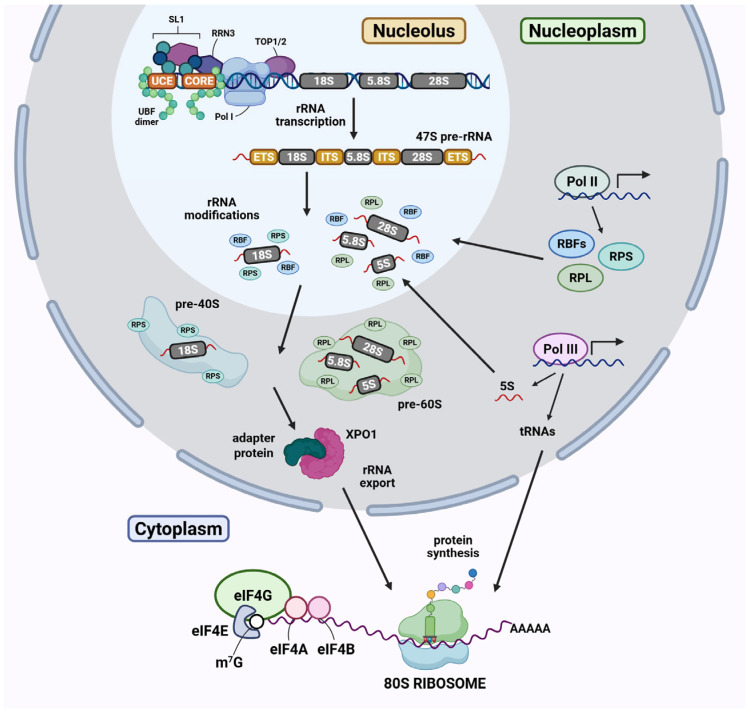
Ribosome biogenesis begins with the rate-limiting step of rRNA gene transcription by Pol I. 47S pre-rRNA transcripts are cleaved, modified, and processed with RPs and RBFs into pre-40S and -60S ribosomes. XPO1 mediates the export of pre-ribosomal subunits into the cytoplasm, where they undergo further modifications and interact with translation factors to form mature 80S ribosomes competent for protein synthesis. (Created in https://BioRender.com).

**Figure 3 cancers-17-02534-f003:**
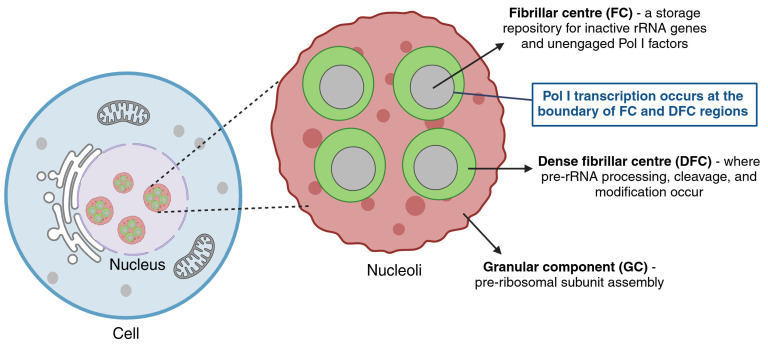
The nucleolus is a membrane-less organelle within the nucleus, where its formation is underpinned by rRNA gene transcription. The fibrillar centre (FC) contains inactive rRNA genes and unengaged Pol I transcription factors and is surrounded by the dense fibrillar centre (DFC) within the granular component (GC). Pol I transcription occurs at the boundary of the FC and DFC. Pre-rRNA processing, cleavage, and modification occur in the DFC. Assembly of pre-ribosomal subunits takes place in the GC. (Created in https://BioRender.com).

**Figure 4 cancers-17-02534-f004:**
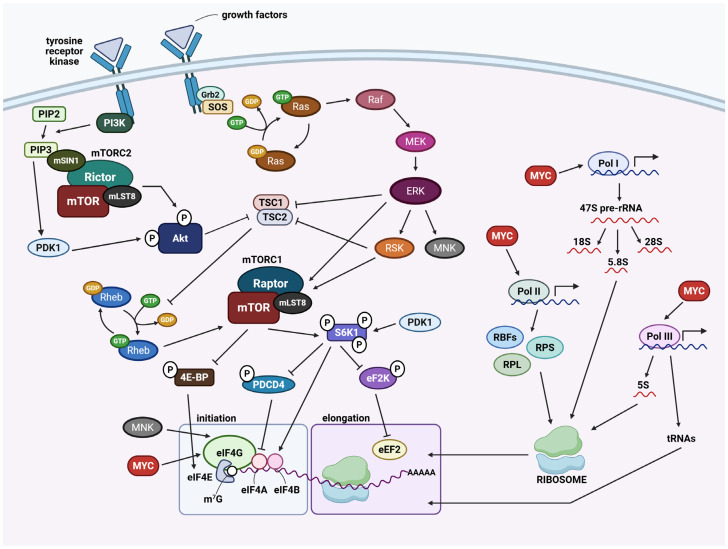
Oncogenic pathways that drive ribosome biogenesis and mRNA translation. Master regulator MYC induces Pol I, II, and III activities, allowing increased production of rRNAs, RPs, RBFs, and other RNA species required for ribosome synthesis and protein production. PI3K/Akt and Ras–ERK pathways, via multiple downstream factors, activate mTORC1, stimulating mRNA translation. (Created in https://BioRender.com).

**Figure 5 cancers-17-02534-f005:**
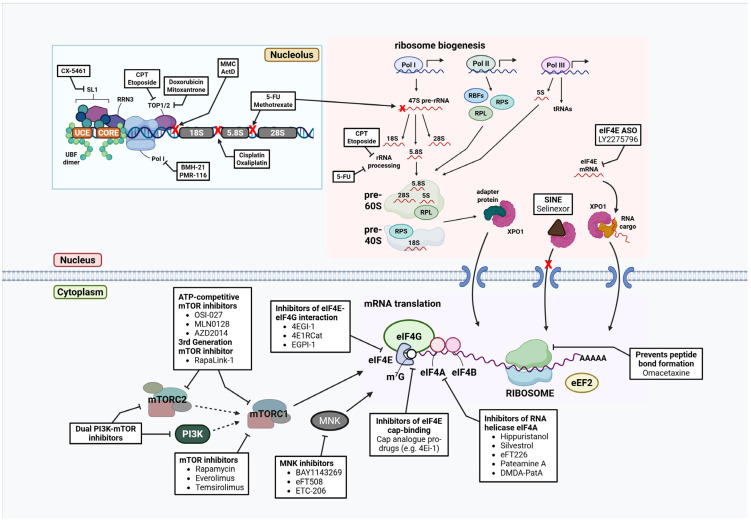
Current landscape of ribosome-targeting strategies in cancer therapy. Chemotherapeutics, including alkylating-like platinum compounds, DNA intercalators, antimetabolites, and topoisomerase inhibitors, inhibit ribosome biogenesis at the rRNA transcription and processing stages. Several Pol I inhibitors (BMH-21, CX-5461, and PMR-116) have been developed to target rRNA gene transcription. Translation initiation is inhibited through direct inhibition of the translational machinery and inhibition of upstream signalling pathways. Omacetaxine blocks translation elongation by preventing the formation of peptide bonds. SINE compound Selinexor (KPT-330) limits nuclear export of various nucleic acids and proteins involved in protein synthesis. (Created in https://BioRender.com).

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
