# Peer review of "Ribosome Biogenesis and Function in Cancer: From Mechanisms to Therapy"

_cancers, 2025, doi:10.3390/cancers17152534_

Round 1
Reviewer 1 Report
Comments and Suggestions for Authors
Gitareja et al. review how ribosomes and ribosome biogenesis can be targeted in cancer therapy. While this topic has already been addressed—for example, by Zisi et al. (https://www.mdpi.com/2072-6694/14/9/2126)—the authors broaden the scope by including additional topics.
Main Comments:
- The authors discuss Diamond-Blackfan anaemia (DBA), which is typically caused by mutations in genes encoding ribosomal proteins of both the small and large subunits. Interestingly, a significant proportion of cases respond to corticosteroid therapy, and some patients experience spontaneous remission in early adulthood. This section would benefit from an explanation of the possible mechanisms through which corticosteroid therapy promotes remission.
- The review mentions promising therapeutic strategies, such as antisense oligonucleotides targeting eIF4E (p. 14). A relevant phase I trial was conducted nearly a decade ago (2016). It would strengthen the manuscript to discuss whether any follow-up trials have been conducted or why further clinical development has not progressed. More generally, the review would benefit from a more comprehensive discussion of recent advances in the field.
Minor Comments:
In the introduction, the authors omit the role of RNA polymerase III, which may cause confusion (even though Pol III is discussed later). The sentence currently reads:
“The production of eukaryotic ribosomes is one of the most intricate biological processes, which relies on the precise coordination of RNA polymerase I (Pol I) transcription of ribosomal RNA (rRNA) genes with various ribosomal proteins (RPs), ribosome biogenesis factors (RBFs), and small nucleolar RNAs (snoRNAs).”
It would be helpful to briefly acknowledge the role of Pol III here to avoid giving the impression that it plays no part in ribosome biogenesis.
Author Response
Reviewer 1
Gitareja et al. review how ribosomes and ribosome biogenesis can be targeted in cancer therapy. While this topic has already been addressed—for example, by Zisi et al. (https://www.mdpi.com/2072-6694/14/9/2126)—the authors broaden the scope by including additional topics.
We appreciate the reviewer’s comment. While there is some overlap with previous reviews, particularly in the discussion of conventional chemotherapies and Pol I transcription inhibitors, our review provides a broader and more updated perspective. Specifically, we focus on therapeutic strategies targeting both ribosome biogenesis and mRNA translation. To distinguish our work, we have expanded the scope to include dedicated sections on direct inhibitors of the translation machinery as well as selective nuclear export inhibitors, thereby offering a more comprehensive overview of current and emerging approaches to targeting the ribosome in cancer therapy.
Main Comments:
- The authors discuss Diamond-Blackfan anaemia (DBA), which is typically caused by mutations in genes encoding ribosomal proteins of both the small and large subunits. Interestingly, a significant proportion of cases respond to corticosteroid therapy, and some patients experience spontaneous remission in early adulthood. This section would benefit from an explanation of the possible mechanisms through which corticosteroid therapy promotes remission.
We have updated the review with a discussion of the possible mechanisms through which corticosteroid therapy promotes remission of DBA as below.
About 80% of DBA cases initially respond to corticosteroid therapy. Some individuals achieve spontaneous remission, defined as having an adequate haemoglobin level for at least 6 months without treatment, by early adulthood. A widely perceived mechanism through which corticosteroids stimulate red blood cell production in DBA is induction of stress erythropoiesis, involving an expansion of erythroid progenitors through activation of glucocorticoid receptor. Studies also indicated that corticosteroids contribute to red blood cell development by modulating signalling pathways, like p53, MYC, and mTOR. However, corticosteroid response in DBA remains elusive due to the complexity of the disease and the multifaceted effects of corticosteroids.
This discussion is added in section 4.1.
- The review mentions promising therapeutic strategies, such as antisense oligonucleotides targeting eIF4E (p. 14). A relevant phase I trial was conducted nearly a decade ago (2016). It would strengthen the manuscript to discuss whether any follow-up trials have been conducted or why further clinical development has not progressed. More generally, the review would benefit from a more comprehensive discussion of recent advances in the field.
We thank the reviewer for this insightful comment. In response, we have updated the section on antisense oligonucleotides targeting eIF4E (Section 5.3.3) to include details regarding the follow-up clinical trial of LY2275796. Additionally, we have substantially expanded the review to include direct inhibitors of the translation machinery and incorporated recent advances in this area. These revisions are included in the section 5.3.3.
Minor Comments:
In the introduction, the authors omit the role of RNA polymerase III, which may cause confusion (even though Pol III is discussed later). The sentence currently reads:
“The production of eukaryotic ribosomes is one of the most intricate biological processes, which relies on the precise coordination of RNA polymerase I (Pol I) transcription of ribosomal RNA (rRNA) genes with various ribosomal proteins (RPs), ribosome biogenesis factors (RBFs), and small nucleolar RNAs (snoRNAs).”
It would be helpful to briefly acknowledge the role of Pol III here to avoid giving the impression that it plays no part in ribosome biogenesis.
We have revised this sentence.
The production of eukaryotic ribosomes is one of the most intricate biological processes, requiring the precise coordination of all three RNA polymerases (Pol I, Pol II and Pol III) alongside numerous auxiliary factors to ensure accurate transcription and synthesis of ribosomal components and assembly of pre-ribosomal particles.
Reviewer 2 Report
Comments and Suggestions for Authors
This article systematically expounds the mechanism of action and targeting strategies of ribosomes in cancer treatment, which has significant frontier value in the discipline. However, there is a significant imbalance in the content structure, and the depth of the drug development section is insufficient.
- The title of this review is "Targeting the Ribosome in Cancer Therapy". Therefore, the focus of the review should be on the types and mechanisms of tumor treatment drugs targeting ribosomes. However, more than half of the descriptions in the full text are discussing the mechanism by which ribosomes function in the body, which is overly redundant.
- The author's description of current anti-tumor drugs is overly simplistic. In the description section of chemotherapeutics (5.1. Existing chemotherapeutics affect ribosome synthesis), the authors mentioned that alkylating agents, DNA embedding agents, antimetabolites and topoisomerase inhibitors can all inhibit ribosome production and exert anti-tumor effects. Then the author should describe their mechanisms of action in detail and attach corresponding diagrams, rather than briefly mention each drug in a few paragraphs and display the mechanisms of action of all drugs in one diagram.
- In 5.2 Selective Ribosome-Targeting agents, the author should also inform readers of why Selective ribosome targeting agents are being developed, what problems current drugs cannot solve, and the current development progress.
- Are selective ribosome-targeting inhibitors only aimed at selectively inhibiting the Pol I target? Are there any other drugs that inhibit the target?
- In the subsequent parts of Chapter Five, the author also needs to rewrite and revise based on the above suggestions.
- The treatment of cancer cannot do without the issue of drug resistance. The author needs to specifically describe in the text the drug resistance problems that may occur during the treatment of tumors with ribosomal drugs and the corresponding mechanisms of drug resistance.
- The author's 6. Ribosome Targeting Strategies: Are Challenges and Promises aimed at ribosome targeting strategies? What I understand is that this part of the outlook is only applicable to 5.2 Selective ribosome-targeting agents. This is not suitable as the conclusion of this review. At the same time, as a review, a simple summary of the future outlook is too hasty.
Author Response
Reviewer 2
This article systematically expounds the mechanism of action and targeting strategies of ribosomes in cancer treatment, which has significant frontier value in the discipline. However, there is a significant imbalance in the content structure, and the depth of the drug development section is insufficient.
- The title of this review is "Targeting the Ribosome in Cancer Therapy". Therefore, the focus of the review should be on the types and mechanisms of tumor treatment drugs targeting ribosomes. However, more than half of the descriptions in the full text are discussing the mechanism by which ribosomes function in the body, which is overly redundant.
We appreciate the reviewer’s feedback. In this review, we aimed to summarize current understanding of ribosome biogenesis and mRNA translation, and how these processes are dysregulated in cancer, as this provides essential background for understanding the rationale, mechanisms of action, and challenges associated with targeting the ribosome in cancer therapy. In response to the reviewer’s comment, we have revised and condensed the background sections to improve focus and clarity, and we have expanded the therapeutic sections to more comprehensively cover the mechanisms of action, clinical progress, and challenges of ribosome-targeting agents. These revisions are highlighted in the manuscript.
- The author's description of current anti-tumor drugs is overly simplistic. In the description section of chemotherapeutics (5.1. Existing chemotherapeutics affect ribosome synthesis), the authors mentioned that alkylating agents, DNA embedding agents, antimetabolites and topoisomerase inhibitors can all inhibit ribosome production and exert anti-tumor effects. Then the author should describe their mechanisms of action in detail and attach corresponding diagrams, rather than briefly mention each drug in a few paragraphs and display the mechanisms of action of all drugs in one diagram.
We appreciate the reviewer’s comment. While many chemotherapeutic agents have been shown to affect ribosome biogenesis, these effects are not necessarily the primary mechanism driving cytotoxicity. In many cases, it remains unclear whether inhibition of ribosome biogenesis directly contributes to cell death. Nonetheless, we acknowledge the importance of providing a more comprehensive overview of how conventional chemotherapeutics impact different steps of ribosome biogenesis. In response, we have revised the relevant section and updated the references to better summarize the current understanding of these mechanisms and have updated Figure 5 to more clearly illustrate how various classes of chemotherapeutic agents target distinct stages of ribosome biogenesis.
- In 5.2 Selective Ribosome-Targeting agents, the author should also inform readers of why Selective ribosome targeting agents are being developed, what problems current drugs cannot solve, and the current development progress.
We thank the reviewer for this valuable comment. To more accurately reflect the mode of action of the compounds discussed, we have revised the title of Section 5.2 to “Pol I Transcription Inhibitors.” In this section, we have expanded our discussion to include a clearer rationale for the development of selective Pol I inhibitors. We have also provided a more comprehensive overview of preclinical studies on drug resistance mechanisms and combination strategies to enhance efficacy. Additionally, we have added a concise paragraph at the end of the section 5.2 summarizing the current clinical development landscape and outlining key challenges, including safety, resistance, and biomarker-based patient selection.
- Are selective ribosome-targeting inhibitors only aimed at selectively inhibiting the Pol I target? Are there any other drugs that inhibit the target?
While direct inhibitors on several ribosome biogenesis proteins such as FBL, NPM1 and PeBoW complex are still in early development, Pol I transcription remains the most advanced and validated therapeutic approach. To make a precise description, the title of section 5.2 has been changed to Pol I transcription inhibitors.
- In the subsequent parts of Chapter Five, the author also needs to rewrite and revise based on the above suggestions.
We have undertaken substantial revisions throughout Chapter Five to align with the earlier comments.
In Section 5.3, “Direct Inhibitors of the Translation Machinery,” we have significantly expanded the content to provide a more comprehensive overview of drug development efforts, progresses and challenges in clinical translation. We also added a summary paragraph at the end of the section 5.3 to highlight key progress and remaining hurdles in this area.
In Section 5.4, “Inhibitors of Signalling Pathways Regulating Translation,” we updated and extended the discussion to include detailed mechanisms of action, resistance pathways, and the issues of clinical development for relevant therapeutic agents.
In Section 5.5, we elaborated on the clinical challenges associated with Selinexor, including treatment-related adverse events, mechanisms of drug resistance, the development of predictive biomarkers, and rational combination strategies to enhance efficacy.
All revisions are highlighted in the manuscript.
- The treatment of cancer cannot do without the issue of drug resistance. The author needs to specifically describe in the text the drug resistance problems that may occur during the treatment of tumors with ribosomal drugs and the corresponding mechanisms of drug resistance.
We thank the reviewer for their constructive suggestion. In response, we have expanded the discussion of resistance mechanisms associated with CX-5461 (section 5.2), inhibitors of translation machinery (section 5.3), inhibitors of translation regulatory pathways (section 5.4) and Selinexor (section 5.4).
- The author's 6. Ribosome Targeting Strategies: Are Challenges and Promises aimed at ribosome targeting strategies? What I understand is that this part of the outlook is only applicable to 5.2 Selective ribosome-targeting agents. This is not suitable as the conclusion of this review. At the same time, as a review, a simple summary of the future outlook is too hasty.
We thank the reviewer for the thoughtful comment and agree that the original Section 6 was not appropriate as the conclusion of the review. In response, we have removed this section. Instead, we have incorporated summary paragraphs within Sections 5.1 to 5.5 to highlight the rationale, current progress, and key challenges associated with each class of therapeutic agents targeting ribosome biogenesis and translation. Additionally, we have revised the Conclusion section to provide an integrated overview of the current understanding of deregulated ribosome biogenesis and mRNA translation in cancer, along with a discussion on therapeutic strategies targeting these vulnerabilities. We believe this revised structure offers a more cohesive and informative conclusion that aligns better with the overall scope and aims of the review.
Round 2
Reviewer 2 Report
Comments and Suggestions for Authors
The author has made thorough revisions to the manuscript in accordance with my suggestions and has reasonably answered my questions. Therefore, I believe this article is now worthy of publication on cancers.